# Pan-Cancer Analysis of Patient Tumor Single-Cell Transcriptomes Identifies Promising Selective and Safe Chimeric Antigen Receptor Targets in Head and Neck Cancer

**DOI:** 10.3390/cancers15194885

**Published:** 2023-10-08

**Authors:** Sanna Madan, Sanju Sinha, Tiangen Chang, J. Silvio Gutkind, Ezra E. W. Cohen, Alejandro A. Schäffer, Eytan Ruppin

**Affiliations:** 1Cancer Data Science Laboratory, National Cancer Institute, Bethesda, MD 20892, USA; sanna.madan@nih.gov (S.M.); ssinha@sbpdiscovery.org (S.S.); tiangen.chang@nih.gov (T.C.); 2Department of Computer Science, University of Maryland, College Park, MD 20742, USA; 3Moores Cancer Center, University of California San Diego, La Jolla, CA 92093, USA; sgutkind@health.ucsd.edu (J.S.G.); ecohen@health.ucsd.edu (E.E.W.C.); 4Department of Pharmacology, University of California San Diego, La Jolla, CA 92093, USA; 5Department of Medicine, University of California San Diego, La Jolla, CA 92037, USA

**Keywords:** CAR-T, single-cell RNA sequencing, immunotherapy, head and neck cancer, glioblastoma, immunoreceptors, cell surface targets, solid tumors, clinical trials

## Abstract

**Simple Summary:**

A major barrier to success for chimeric antigen receptor (CAR) T-cell therapies in solid tumors is selecting optimal target antigens. Target antigens should be uniformly expressed by cancer cells and minimally so by healthy tissues in the body. Analyzing single-cell RNA-sequencing data of patient tumors and of healthy tissue reference atlases, we first charted the landscape of existing solid tumor CAR targets, quantifying their tumor selectivity and safety to identify the leading clinical targets. We then compared these benchmark targets to the cell surfaceome, with the aim of systematically identifying new candidate genes that are safer and/or more selective than the best CAR targets in clinical trials, as determined by our measures. Our analysis uncovered twenty new cell surface targets, five of which have both superior selectivity and safety scores, which may serve as promising candidates for CAR treatments in HNSC.

**Abstract:**

Chimeric antigen receptor (CAR) T cell therapies have yielded transformative clinical successes for patients with blood tumors, but their full potential remains to be unleashed against solid tumors. One challenge is finding *selective* targets, which we define intuitively to be cell surface proteins that are expressed widely by cancer cells but minimally by healthy cells in the tumor microenvironment and other normal tissues. Analyzing patient tumor single-cell transcriptomics data, we first defined and quantified selectivity and safety scores of existing CAR targets for indications in which they are in clinical trials or approved. We then sought new candidate cell surface CAR targets that have better selectivity and safety scores than those currently being tested. Remarkably, in almost all cancer types, we could not find such better targets, testifying to the near optimality of the current target space. However, in human papillomavirus (HPV)-negative head and neck squamous cell carcinoma (HNSC), for which there is currently a dearth of existing CAR targets, we identified a total of twenty candidate novel CAR targets, five of which have both superior selectivity and safety scores. These newly identified cell surface targets lay a basis for future investigations that may lead to better CAR treatments in HNSC.

## 1. Introduction

Chimeric antigen receptor T cell (CAR-T) therapy has revolutionized cancer treatment. After a patient’s T cells are extracted, they are modified to recognize a tumor-specific protein via the addition of a synthetic CAR on the cell membrane and then infused back into the patient. CAR-T treatment has resulted in durable remissions in patients, particularly in those with B-cell malignancies. Recently, CARs directed against the B-cell marker CD19 have been declared curative, with observed remission rates as high as 90% [1]. The physiological phenomenon that makes B-cell tumors more amenable to CAR-T therapy than other tumors is that one can kill all the B-cells—both cancerous and not cancerous—and the patient is likely to survive, especially if treated with intravenous immunoglobulin (IVIG) to replace the missing antibodies.

As of August 2023, the Trialtrove repository of clinical trials [2] had recorded 2119 clinical trials involving cells engineered to target various membrane surface proteins or other antigens in specific indications that have either taken place, are currently in progress, or are proposed/planned; 482 of these trials are of phase II or higher. Most of these trials involve modified CAR T cells, and some involve NK cells; because our analysis applies to both T and NK cells, we use “CAR” without a suffix in most places. The CAR targets that are currently Food and Drug Administration (FDA)-approved or are being tested include a variety of proteins, e.g., CD19 and CD22 in B-cell malignancies, and HER2, IL-13 receptor α2, EGFRvIII, carbonic anhydrase IX, and MUC1 in various solid tumors [3].

While CAR-T treatments have been remarkably successful against B-cell tumors, several challenges remain in realizing their potential in the realm of solid tumors. Two key challenges are (1) heterogeneity of cell surface proteins expressed by a tumor’s constituent malignant cells and (2) expression of the CAR target protein on the cell surfaces of normal tissues elsewhere in the body, posing a toxicity risk. These challenges may be addressed by studying single-cell RNA sequencing (scRNA-seq), which has generated many patient tumor transcriptomes in recent years. We hypothesized that we could harness these single-cell measurements to profile the expression of genes encoding targets of CAR therapies and, more broadly, cell surface proteins at a cellular resolution—thus enabling us to identify suitable surface proteins that have not previously been considered seriously as targets for CAR therapy. Indeed, recent studies have harnessed scRNA-seq to investigate the toxicities of CAR-T therapies [4,5] and highlighted the importance of optimizing surface target selection by utilizing scRNA-seq [6,7,8]. Furthermore, a notable recent analysis leveraged public patient tumor and reference atlas scRNA-seq data to first construct an integrated “meta-atlas” of tumor and normal cells and then identify tumor-specific combinations of target antigens via logic gates [9]. Hopefully, identifying optimal cell surface targets in a systematic, data-driven manner may translate in the future to improved patient outcomes, leading to more durable remissions and reduced toxicities.

Zeroing in on the challenges listed above, we asked whether mining scRNA-seq datasets of solid tumors can enable us to perform the following: (1) First, chart the landscape of existing cell-surface targets at which CARs are currently aimed in clinical trials and estimate their selectivity and safety with quantitative scores. (2) Second, identify new, alternative target proteins that are differentially highly expressed by tumor cells and lowly expressed by non-tumor cells within the tumor microenvironment (TME) and additionally in normal tissues across the body. Our goal is to identify new targets for CAR therapies that have better safety and selectivity scores than the targets currently tested in an indication-specific manner.

We use the term “indication” to refer to an ordered pair of (cancer type, CAR target) that has been proposed in at least one clinical trial; we use the opposite term “non-indication” to denote a (cancer type, CAR target) for which we could not find any proposed clinical trial (Methods). Our analysis proceeds in the following steps: (1) First, we mine clinical trial databases and the literature, and we identify the current CAR-T targets in clinics mapped to the solid tumors to which they are indicated/targeted. For each (cancer type, CAR target) pair, we compute a “tumor selectivity score”, i.e., the selectivity of the expression of a given candidate target gene in tumor cells versus all other cells in the TME. This is quantified by the ability of a classifier to separate tumor cells from non-tumor cells based on the single-cell expression data of this gene in the TME of the respective cancer indication. (2) Second, we compute a “safety score” for each candidate target, which is computed by quantifying the level of protein and scRNA-seq expression across normal human tissues. We additionally investigate the in vitro essentiality of genes encoding CAR targets that are either approved or being tested in clinics, as well as the candidate new targets yielded by our analysis in CRISPR knockout screens, thus evaluating a “dependency score”. The overview of our approach showing all three resource inputs is shown in the Graphical Abstract.

The first part of our analysis evaluates the indications that are currently studied in clinical trials (or already approved) for the tumor selectivity and safety metrics of their pertaining targets. We then compare new candidate CAR target surface protein versus these current state-of-the-art benchmarks. Our goal is to identify new target genes that match or exceed the leading extant targets’ selectivity and safety scores in each respective cancer type, thus forming promising new target tumor antigens for further possible consideration when engineering new CARs. Differently from previous related studies [7,9], the unique aim of our study is to find novel targets for CARs that are strictly better than the best CAR target antigens in the clinics based on selectivity and safety metrics. This analysis, quite surprisingly, identifies only a few but highly promising candidates for clinical translation.

## 2. Materials and Methods

### 2.1. Clinical Trial Curation

To obtain a list of CAR clinical trials and their targets in solid tumors, we first obtained a list of single CAR targets in clinical trials from MacKay et al., who found such a list by carrying out a ClinicalTrials.gov query on 3 March 2019 [10] (accessed on 21 August 2021). To obtain a larger and more up-to-date list of clinical trials since then, an additional search was carried out on ClinicalTrials.gov on 20 August 2021 for CAR-T clinical trials posted on 3 March 2019 and after. The following query was used: “CAR-T OR CAR OR chimeric antigen receptor OR chimeric immunoreceptors OR artificial T-cell receptors|Recruiting, Not yet recruiting, Active, not recruiting, Completed, Enrolling by invitation, Suspended, Terminated Studies|Interventional Studies|(cancer OR carcinoma OR solid tumor OR solid tumors OR melanoma OR sarcoma) NOT (hematological OR lymphoid OR liquid cancers OR lymphoma OR leukemia OR CD19 OR B cell malignancy OR myeloma OR Hematologic OR blood cancer)|First posted from 3 March 2019 to 20 August 2021”. CAR trials and targets were further identified through manual curation, and clinical trial statuses were cross-checked in TrialTrove (https://citeline.informa.com/trials/results) to ensure consistency [2] (first checked on 10 September 2021 and again on 16 August 2023). Access to TrialTrove requires a license.

CAR protein targets were then selected for further analyses if they had a corresponding single gene that could be analyzed for gene expression. The following molecules were left out of the analysis: Claudin 18.2, CD44v6, EGFRvIII, k-IgG, LeY, CEA, and GD2 because (i) they are not proteins or (ii) they are protein isoforms whose presence and abundance cannot easily be estimated from single-cell gene expression data. For each target, its indications were identified as the specific tumor types that were mentioned in its ClinicalTrials.gov records, excluding “solid tumors” and “metastases”. The following mapping of clinical trial cancer types to our datasets was used: brain cancer or glioma → glioblastoma; pancreatic cancer → pancreatic adenocarcinoma; stomach cancer or gastric cancer → stomach adenocarcinoma; liver or hepatocellular carcinoma → liver hepatocellular carcinoma; head and neck cancer → head and neck squamous cell carcinoma; melanoma → skin cutaneous melanoma; colon cancer or colorectal cancer → colorectal cancer; ovarian cancer → ovarian serous cystadenocarcinoma; lung cancer or non-small cell lung cancer (but not small cell lung cancer) → non-small cell lung cancer; and breast cancer → breast invasive carcinoma. Liquid tumor CAR targets were taken from MacKay et al. [10].

### 2.2. Single-Cell Transcriptomics Data Collection

We downloaded single-cell transcriptomics datasets from the Tumor Immune Single-cell Hub (TISCH) 1.0 database [11]. This resource contains 76 uniformly processed scRNA-seq datasets spanning 28 cancer types and containing data from over 2 million cells. We selected datasets from human tissues that contained both malignant and non-malignant cells. This yielded a total of 26 unique solid tumor datasets: two of breast cancer (BRCA), one of colorectal cancer (CRC), eleven of glioblastoma (GBM), one of head and neck squamous cell carcinoma (HNSC), one of liver hepatocellular carcinoma (LIHC), four of non-small cell lung cancer (NSCLC), one of ovarian cancer (OV), two of pancreatic adenocarcinoma (PAAD), two of skin cutaneous melanoma (SKCM), and one of stomach adenocarcinoma (STAD). These data were all uniformly processed by TISCH as log_2_ (transcripts per 10,000 counts in each cell + 1).

### 2.3. Single-Cell Data Analysis

In-house programs written in Python 3 were used to carry out analyses on the scRNA-seq datasets. A cell was labeled “tumor” if it was labeled “malignant” by TISCH and “non-tumor” otherwise. We then partitioned the cells in each dataset by patient (if such annotation was available); the partition by patient is so that each patient counts equally in the downstream analysis regardless of how many cells were sampled from that patient. As a quality control, only patient samples that contained at least 25 tumor cells and 25 non-tumor cells were selected for analysis. After this filtering step, only the cancer types where we had samples from at least three remaining patients were subsequently considered, excluding breast cancer entirely from our analysis, for which we only had two patients.

A list of 3559 cell surface protein-coding genes was obtained from Hu et al. [7]; subsequently, we use the term “surfaceome” to refer to these genes or the proteins they encode. Genes whose symbols differed in that reference in TISCH and in DepMap (see later subsection) were resolved by using the Human Gene Nomenclature Committee table, including gene aliases (downloaded from https://www.genenames.org) (accessed on 3 February 2022). We generated patient-specific AUCs (AUC: area under the receiver operating characteristic (ROC) curve) of tumor-vs-non-tumor cell discrimination for each gene whose respective protein was listed in a CAR clinical trial and for all cell surface protein-encoding genes at large.

Tumor selectivity scores (i.e., AUCs) were computed as follows: for a given cell surface gene A, there is an array containing its expression across all cells measured in a given patient: *expression_levels*. There is a corresponding array indicating whether each respective cell is a tumor cell (“1”) or a nontumor cell (“0”): *cell_labels*. The AUC is then computed in Python as roc_auc_score (*cell_labels*, *expression_levels*), where roc_auc_score is a function from sklearn.metrics. A higher AUC indicates higher expression in tumor cells compared to nontumor cells in the patient’s sample.

For each existing (CAR target, indication), we compute a mean tumor selectivity score over all patient-specific selectivity scores (i.e., AUCs). We took the top-ranking target in each cancer type by the mean tumor selectivity scores as the threshold to which we compared all other cell surface protein-encoding genes. In making these comparisons, we used the Wilcoxon signed rank one-sided test comparing matched patient-level AUCs with Benjamini–Hochberg false discovery rate (FDR) *p*-value < 0.1, denoting statistical significance, repeating this procedure for each cancer type separately.

### 2.4. Reference Atlas Analysis: Computing Safety Scores

For computing the safety score of a cell surface protein in the Human Protein Atlas (HPA), an approach similar to that outlined in MacKay et al. [10] was taken. Individual tissue types were grouped into tissue groups with the following mapping (Table 1):

HPA protein measurements for a given tissue type are either “Not detected”, “Low”, “Medium”, or “High”. For a tissue group, if the measurements of a protein over all tissue types only ranged from “Not detected” to “Low”, then the tissue group was assigned “ND-L” (not detected or low). If the measurements over all tissue types only ranged from “Medium” to “High”, then the group was assigned “M-H” (medium to high). If values from either “Not detected” or “Low” and either “Medium” or “High” were present in the measurements over all tissue types, then the group was assigned “Variable”. Thus, in determining the overall HPA safety score of a protein, the scoring weight was in the following order: ND-L > Variable > M-H. For each tissue group, a score of 2 was added for each “ND-L”, 1 was added for each “Variable”, and 0 was added for each “M-H”. The mean of the resulting sum was taken to obtain a representative measure over all tissue groups that were measured for the protein. As such, the scoring values ranged from 0 to 2.

For the Tabula Sapiens safety score computation, the tissues measured were taken as is, with no further grouping, as the following: bladder, blood, bone marrow, eye, fat, heart, kidney, large intestine, liver, lung, lymph node, mammary gland, muscle, pancreas, prostate, salivary gland, skin, small intestine, spleen, thymus, tongue, trachea, uterus, and vasculature. To compute the safety score of a surfaceome gene, first, within each tissue, the percentage of cells expressing that gene was computed. Next, for a tissue, if the percentage was < 1%, 10 was added to the final score; if it fell between 1 and 10%, 9 was added to the final score; if it fell between 11 and 20%, 8 was added to the final score; if it fell between 21 and 30%, 7 was added to the final score; if it fell between 31 and 40%, 6 was added to the final score; if it fell between 41 and 50%, 5 was added to the final score; if it fell between 51 and 60%, 4 was added to the final score; if it fell between 61 and 70%, 3 was added to the final score; if it fell between 71 and 80%, 2 was added to the final score; if it fell between 81 and 90%, 1 was added to the final score; and finally, if it fell between 91 and 100%, 0 was added to the final score. The mean of this sum was taken over the total number of tissue groups for which the gene had been measured. As such, the scoring values ranged from 0 to 10.

### 2.5. DepMap Essentiality (Dependency) Scores

We used the DepMap Public 22Q2, specifically “CRISPR_gene_dependency.csv”, which was downloaded from https://depmap.org/portal/. This file contains the probability of dependence scores for each (cell line, gene) pair. The DepMap defines a probability of ≥ 0.5 as essential, and as such, our analysis defined a cell line as dependent on a gene if its p(dependent) ≥ 0.5.

Cell lines were partitioned by cancer type for each clinical or novel CAR target gene. For CRC, we analyzed cell lines labeled “Colon/Colorectal Cancer” as the primary disease; for GBM, we analyzed cell lines labeled “Brain Cancer” as primary disease and “Glioblastoma” as the subtype; for NSCLC, we analyzed cell lines labeled “Lung Cancer” as primary disease and “Non-Small Cell Lung Cancer (NSCLC)” as the subtype; for PAAD we analyzed cell lines labeled “Pancreatic cancer” as the primary disease and “Adenocarcinoma” as the subtype; for STAD we analyzed cell lines labeled “Gastric Cancer” as the primary disease and “Adenocarcinoma” as the subtype; for OV we analyzed cell lines labeled “Ovarian Cancer” as the primary disease with no constraint on subtype; for SKCM we analyzed cell lines labeled “Skin Cancer” with “Melanoma” as the subtype; for HNSC we analyzed cell lines labeled “Head and Neck Cancer” as the primary disease and “Squamous Cell Carcinoma” as the subtype; for B-cell Non-Hodgkin’s Lymphoma we analyzed cell lines labeled “Lymphoma” as the primary disease and “B-cell, Non-Hodgkin’s” as the subtype; for B-cell Hodgkin’s Lymphoma we analyzed the cell lines as primary “Lymphoma” and the subtype as “B-cell, Hodgkin’s”; for T-cell, Non-Hodgkin’s Lymphoma, we analyzed cells labeled “Lymphoma” as the primary disease and “T-cell, Non-Hodgkin’s” as the subtype; for T-cell Lymphoma we analyzed cell lines with “Lymphoma” as the primary and “T-cell” as the subtype; for B-cell Lymphoma, we analyzed cell lines with “Lymphoma” primary and “B-cell” subtype; for Acute Myelogenous Leukemia (AML) we analyzed cell lines with “Leukemia” primary and “Acute Myelogenous Leukemia” in the subtype; for Chronic Myelogenous Leukemia we analyzed cell lines with “Leukemia” as the primary and “Chronic Myelogenous Leukemia” as the subtype; for B-cell Chronic Lymphoblastic Leukemia (B-CLL), we analyzed cell lines with “Leukemia” as the primary and “Chronic Lymphoblastic Leukemia” as the subtype; for B-cell Acute Myeloid Leukemia (AML) we analyzed cell lines with “Leukemia” as the primary and “Acute Lymphoblastic Leukemia (ALL), B-cell” as the subtype; for T-cell AML, we analyzed cell lines with “Leukemia” as the primary and “Acute Lymphoblastic Leukemia (ALL), T-cell” as the subtype; for unspecified B-cell Leukemia we analyzed cell lines with “Leukemia” as the primary and “B-cell, unspecified” as the subtype; and for Myeloma we analyzed cell lines with “Myeloma” as the primary.

For each (cancer type, gene) indication, the mean p(essentiality) of the gene was simply taken over all p(essential) values in the cell lines of the cancer type indication as mapped above. The DepMap (https://depmap.org/portal/documentation/) defines a “strongly selective” gene “as having a dependency distribution that is at least 100 times more likely to have been sampled from a skewed t-distribution than a normal distribution”, i.e., its skewed LRT value exceeds 100. A “common essential” (i.e., “pan-essential”) gene “is defined as a gene which, in a large, pan-cancer screen, ranks in the top X most depleting genes in at least 90% of cell lines. X is chosen empirically using the minimum of the distribution of gene ranks in the 90th percentile least depleting lines”.

### 2.6. TCGA Primary Tumor vs. Matched Normal Analysis

HNSC TCGA RNA-seq expression data were downloaded from https://toil-xena-hub.s3.us-east-1.amazonaws.com/download/tcga_RSEM_gene_tpm.gz (accessed on 6 August 2023). The expression units were log_2_(TPM + 0.001), where TPM stands for transcripts per million. Comparisons between primary HNSC tumors and matched adjacent normal tissues were carried out by pairing matching sample IDs between primary tumor and adjacent normal and then performing a one-sided paired t-test for our twenty targets and the two known clinical CAR targets for HNSC.

## 3. Results

### 3.1. Overview of the Analysis

We analyzed nine cancer types that have sufficient data in the scRNA-seq Tumor Immune Single-cell Hub (TISCH) 1.0 collection: colorectal cancer, glioma, (HPV-negative) head and neck cancer, liver cancer, non-small cell lung cancer, ovarian cancer, pancreatic adenocarcinoma, melanoma, and stomach cancer [11] (Methods). To quantify the tumor selectivity of a given gene encoding either an extant or new candidate CAR target cell surface protein, we used the “area under the curve” (AUC) metric, which quantifies the extent by which its expression levels can be used to discriminate between tumor and non-tumor cells in the TME (Methods). We use “AUC” and “tumor selectivity score” interchangeably. The tumor selectivity score is computed for each patient individually, such that for a given cell surface protein target in a cancer type, we obtain a distribution of tumor selectivity scores across all patients, whose mean and standard deviation can be quantified.

In parallel, to quantify the overall safety of a target gene encoding a cell surface protein, a two-part safety score was computed, considering the Human Protein Atlas (HPA) proteomics data [12] and the scRNA-seq Tabula Sapiens (TS) gene expression data separately [13]. Each of these two sub-safety scores is computed at the tissue level and averaged over the set of pertaining samples; for each resource, the safety measures are computed for each tissue separately, and then the arithmetic mean of safety scores across all tissues is taken as the final safety score (Methods). The HPA safety scores range from 0 to 2, and the TS safety scores range from 0 to 10, following each resource’s scoring.

### 3.2. The Quantified Tumor Selectivity and Safety Landscape of the Human Surfaceome

To obtain a birds-eye view of the tumor selectivity and safety metrics outlined in the previous section, we quantified their values across all surfaceome genes (curated from Hu et al. [7]). The cancer type-specific distributions of tumor selectivity scores across all surfaceome genes are shown in Figure 1A. The distributions are highly similar across cancer types: all are symmetric and unimodal, with AUC values tightly concentrated around 0.50.

We examined how many highly tumor-specific (AUC > 0.75) surfaceome genes were upregulated in each cancer type (Figure 1B). Interestingly, the number of such surfaceome genes varies considerably by cancer type, with the lowest values in GBM (*n* = 2), OV (*n* = 6), and LIHC (*n* = 7) and the highest values in SKCM (*n* = 42), CRC (*n* = 50), and HNSC (*n* = 84). This suggests that the latter subset of solid tumor types may contain a larger space of surface targets that may be considered for CAR therapeutic development compared to the former subset of solid tumor types. Finally, the distribution of safety scores derived from the HPA and Tabula Sapiens atlases are shown in Figure 1C,D, respectively. Both distributions are strongly skewed to the left; most surfaceome genes and proteins have considerably high safety scores (HPA safety mean = 1.38, median = 1.55, and standard deviation = 0.53; Tabula Sapiens safety mean = 8.56, median = 9, and standard deviation = 1.54). Additionally, the quantified cancer type-specific selectivity scores and safety scores derived from each resource (with tissue-specific resolution) for all surfaceome genes are provided in Appendix A.

### 3.3. The Selectivity and Safety Landscape of Targets of Approved and Currently Studied CARs

Mining TrialTrove, ClinicalTrials.gov, and the existing literature [10], we assembled a table summarizing the genes encoding proteins that are currently being targeted by CARs in clinical trials in each cancer type, including those already approved (Table 2 and Methods). We term these clinical CAR targets. For almost all the solid tumors we have considered, we find at least 10–20 unique genes encoding cell surface proteins that are currently targeted by CARs in various stages of active or planned clinical trials. The most frequently studied targets among those are PSCA, MUC1, CD274, EGFR, and ERBB2. Notably, however, head and neck cancer (HNSC) has a dearth of unique CAR targets, with only two targets currently in clinical trials, EGFR and ERBB2, suggesting that there may be a critical unmet need to identify additional cell surface targets for HNSC. Following this assessment, we charted the distribution of tumor selectivity scores for each cancer type’s pertaining clinical CAR targets (Table 3, Figure 2A,B).

Next, for each cancer type studied, we identified (1) the leading clinical CAR target based on its tumor selectivity score, as derived from TISCH scRNA-seq data, (2) the leading clinical CAR target based on its safety score, as derived from the Human Protein Atlas (HPA), and (3) the leading clinical CAR target based on its safety score, as derived from the Tabula Sapiens (TS) scRNA-seq reference atlas (Table 3). The scores of the leading targets by each criterion are further used as threshold parameters by which any gene encoding a cell surface protein can be compared to quantify its potential to serve as a CAR target.

Reassuringly, we find that the tumor selectivity scores of genes encoding clinical CAR targets are higher than other surfaceome genes in a statistically significant manner in six cancer types (Wilcoxon rank-sum *p*-value < 0.05) (Figure 2A). Notably, this testifies to the utility of the single-cell tumor selectivity metric we have derived, as this measure is demonstrably higher among genes encoding surface proteins that have already been vetted to be targeted by CARs. We similarly chart the distributions of safety score values, also encouragingly revealing that the safety scores we derived from the HPA are higher among clinical CAR targets than all surfaceome proteins (Figure 2C, Wilcoxon rank-sum *p* = 0.11). However, we note that the median TS-derived safety scores of all surfaceome genes and genes encoding clinical CAR targets are roughly equivalent (Figure 2D, Wilcoxon rank-sum *p* = 0.95). We continue to use this sub-safety score in what follows to adopt a cautious approach and provide as comprehensive a picture as possible, but we note that it is not discriminatory as the tumor selectivity and HPA safety scores are. We additionally plotted the distributions of safety scores in all surfaceome genes and cancer-type-stratified clinical CAR targets (Appendix A), as well as comparisons between genes encoding clinical CAR targets and the most tumor-selective surfaceome genes in each cancer type (Appendix A).

### 3.4. Identifying Novel CAR Targets with Higher Estimated Selectivity and Safety Scores

In the previous subsection, for each cancer type, we identified the top-ranking existing CAR targets based on their tumor selectivity and safety scores. We now pivot to identify new candidate targets that have better selectivity and safety scores than those of the current therapies. We proceeded in three steps: (1) identifying highly selective CAR targets (with yet moderate toxicity levels), (2) identifying high safety-scoring CAR targets (with yet moderate selectivity levels), and finally, (3) identifying the targets that have both very high selectivity and safety scores (Figure 3A). These three steps were conducted for every indication, comparing the scores of the new targets to the best-scoring CAR targets in that cancer type.

We first aimed to identify novel highly selective CAR targets that yet have potentially acceptable low toxicity. We searched for surface proteins that are encoded by genes whose selectivity scores across patients in a cancer type are significantly higher than the respective top currently established CAR target (employing a one-sided Wilcoxon signed rank test comparing patient-level selectivity scores, with FDR *p*-value < 0.1 denoting significance). To exclude targets with potentially extreme toxicity levels and only consider those that are most likely reasonable/safe, we applied lower bound thresholds to each of the sub-safety scores: 1.5 (out of 2) for HPA and 7.5 (out of 10) for Tabula Sapiens. Notably, the only cancer types where we find new higher-scoring selective targets are HPV-negative HNSC (seven targets) and glioblastoma (GBM; one target) (Figure 3B and Appendix A).

Second, studying a complementary perspective, we next searched for highly safe targets, which are those that have higher safety scores than the best approved/studied CAR target in a cancer type. We concomitantly required that these targets are at least moderately selective, with mean tumor selectivity scores ≥ 0.7. We required that new candidate targets’ safety scores surpass both atlases’ safety scores of the pertained cancer type-specific top target(s). Notably, here, we found high-ranking targets only for HPV-negative HNSC. The eighteen newly suggested targets are shown in Figure 3C and Appendix A, along with the established target ERBB2.

Finally, we searched for target genes that have selectivity and safety scores that are both higher than those observed for the current targets studied within each cancer type. We find five such targets, all for HPV-negative HNSC (Figure 3D). Indeed, charting the tumor selectivity scores and HPA safety scores of all surfaceome genes in two dimensions (Tabula Sapiens safety scores were excluded to enable clearer visualization but are provided in Appendix A), along with the cancer type-specific thresholds for both metrics, revealed that HNSC is the only cancer type for which there exist any surface targets that surpass both thresholds and are thus more selective and safer than the leading existing CAR targets (Figure 3A,D).

To complement the safety scores shown in Figure 3, the tissue-specific expression levels at both the proteomic (HPA) and single-cell RNA-sequencing (Tabula Sapiens) levels are shown for all twenty newly uncovered cell surface targets in Appendix A, in conjunction with those of existing solid tumor CAR targets in clinics.

We additionally cross-examined the twenty unique cell surface targets yielded by our analysis in TCGA data [14]. Specifically, we conducted a one-sided paired t-test comparing the expression levels of each of our targets in primary HNSC tumors to the matched adjacent tumor-adjacent tissues. Eighteen of our targets were indeed expressed more highly in tumors than in their corresponding matched normal tissues (FDR-corrected *p*-value < 0.05) (Appendix A). The gene encoding the approved CAR target ERBB2 was expressed more highly in matched normal tissue than in tumor tissue, indicating that the new targets uncovered herewith may possess greater tumor-vs-nontumor stratification abilities than at least one of two established HNSC CAR targets.

### 3.5. Further Ranking Top Predicted CAR Targets by Their Essentiality (Dependency) Scores in Tumor Cells

To evaluate the potential of a surface protein to serve as a CAR target, it is of further interest to study its essentiality in cancer cells. A high essentiality of a target in its pertaining cancer type testifies to a likely dependency on that protein by the tumor cells and, subsequently, points to a possible decreased potential for immune evasion via downregulation of this gene and loss of its corresponding antigen on the cell surface. To this end, we analyzed the CRISPR DepMap [15] essentiality scores for these genes (Figure 4A,B, Appendix A) along with the untested candidate targets yielded by our pipeline (Figure 4C, Appendix A). In this analysis, we considered a cell line dependent on that gene if p(dependence) ≥ 0.50, as defined by DepMap (Methods). For each target gene, we computed the mean p(dependence) over all cancer cell lines of that indication and the fraction of cell lines in which the gene was deemed essential. We also note that DepMap subcategorizes “essential” genes as either “strongly selective” or “common-essential”. The former denotes strong essentiality in a subset of cell lines, and the latter denotes broad, pan-cancer essentiality in almost every cell line.

Intriguingly, most CAR targets in clinical trials exhibit quite low dependency scores across cell lines of their respective indications. One notable exception is the FDA-approved CAR-T target-encoded gene CD19, which is essential in 75% (9/12) of non-Hodgkin’s B-cell lymphoma cell lines, 25% (3/12) of B-cell acute lymphoblastic leukemia (B-ALL) cell lines, and 100% (3/3) of unspecified B-cell leukemia cell lines (Figure 4B, Appendix A). CD19 is also deemed “strongly selective” by DepMap. Interestingly, the other blood tumor clinical CAR-T target genes, CD22 (in Phase 1 trials) and TNFRSF17 (encoding the BCMA protein and FDA-approved against multiple myeloma), are not essential in any cell line of their respective indications analyzed here.

In solid tumors, the emerging overall picture is similar. Almost all clinical CAR targets are not essential in any of the cell lines in the pertaining indications, with the striking exceptions of EGFR and ERBB2, which are highly essential across all their indications. For each cancer type they are indicated in, EGFR and ERBB2 consistently rank the highest both in terms of the fraction of cell lines they are essential in and the mean and median p(dependency) across cell lines (Figure 4A). EGFR is essential in 76% of HNSC cell lines, and ERBB2 is essential in 52% of HNSC cell lines (Appendix A). Out of the 35 solid tumor clinical CAR target genes that we collected and analyzed, EGFR, ERBB2, and MET are the only ones that are deemed essential by the CRISPR DepMap, all “strongly selective”.

Of the candidate CAR targets yielded by our pipeline, the vast majority are not essential in any cell lines of their respective cancer indication. However, two targets stand out: IGSF3 and SLC2A1, which are essential in 80% and 45% of HNSC cell lines, respectively (Figure 4C, Appendix A). IGSF3 is classified as “common-essential” (i.e., “pan-essential”) and SLC2A1 is classified as “strongly selective” by DepMap.

## 4. Discussion

Overall, there are twenty unique genes encoding surface proteins identified by our pipeline across all three filtering configurations (top selective, top safe, and requiring both concomitantly). The twenty genes, the surface proteins they encode, and the families/classes of the respective proteins are shown in Table 4. Here, we present a narrative summary of seven of those that were yielded by our analysis: the five targets that are both highly selective and safe and two that are already existing targets of CAR therapies or antibody–drug conjugates in different indications than HNSC. Additional narratives for the remaining candidate surface targets can be found in Appendix A.

We begin with *TACSTD2*, which has a very high selectivity score in HNSC (mean AUC = 0.94 and one-sided Wilcoxon signed rank test FDR *p*-value = 0.065). It encodes the tumor-associated calcium signal transducer 2, also known as “Trop-2” and “epithelial glycoprotein-1 antigen”, a calcium signal transducing cell surface receptor and known tumor-associated antigen. Among all cancers in The Cancer Genome Atlas (TCGA) [16], HNSC harbors the third highest median *TACSTD2* expression levels (265.2 fragments per kilobase million (FPKM0)). In the HPA Pathology Atlas [17], antibody staining revealed medium-to-high membranous and/or cytoplasmic positivity in two of four HNSC patients. Biallelic loss-of-function mutations in *TACSTD2* are known to cause gelatinous drop-like corneal dystrophy [18], which suggests that the expression of this gene is functionally important only to the eye. Notably, Trop-2 is the target of sacituzumab govitecan [19], an antibody–drug conjugate that is FDA-approved to treat metastatic triple-negative breast, HR-positive breast, and urothelial cancers. The existence of this approved drug may provide a basis for expedited anti-Trop2 CAR development for patients with HNSC and other solid tumors.

Next, *DSG3* has both improved tumor selectivity (AUC = 0.93 and FDR *p* = 0.065) *and* safety (Tabula Sapiens score = 9.13 and HPA score = 1.79) over the leading extant targets in HNSC. This gene encodes the transmembrane glycoprotein desmoglein-3, which belongs to both the desmoglein family and the cadherin cell adhesion molecule group. *DSG3* median gene expression levels were the highest in HNSC among all TCGA cancer types (median = 155 FPKM), and its encoded protein was detected in two of four HNSC patients via membranous and/or cytoplasmic antibody staining in the HPA Pathology Atlas. Additionally, *DSG3* was previously found to be overexpressed at both the RNA and protein levels in head and neck cancer [20]. Inactivating mutations in the *DSG3* gene can lead to the formation of acantholytic blisters in the oral and laryngeal mucosa [21]. Importantly, the autoimmune disease pemphigus vulgaris is caused by autoantibodies targeting desmoglein-3 [22], resulting in blistering and erosions of the skin and mucosae (accordingly, desmoglein-3 is also known as “pemphigus vulgaris antigen”). Given this critical caveat, we caution that developing CARs against all DSG3-expressing cells is likely risky and that a better strategy is likely to target only DSG3-overexpressing cells.

*CDH3* also has higher clinical tumor selectivity and safety scores in HNSC than existing targets (AUC = 0.92 and FDR *p*-value = 0.065; TS safety = 9.21; and HPA safety = 1.58). It encodes the cadherin-3 protein, which is a member of the cadherin superfamily, cell adhesion proteins that are dependent on calcium and enable cells to stick together by binding to each other. *CDH3* exhibited the highest median RNA expression in HNSC among all cancer types in the TCGA (median = 90.9 FPKM). Moreover, HPA Pathology antibody staining revealed all four out of four HNSC patients exhibited medium/high CDH3 protein expression. Previous studies have reported *CDH3* overexpression and association with poor prognosis in HNSC [23,24,25]. Biallelic mutations in *CDH3* can cause recessive congenital hypotrichosis with juvenile macular dystrophy [26], a rare disorder characterized by sparse hair and progressive vision loss, as well as ectodermal dysplasia, ectrodactyly, and macular dystrophy syndrome, similar to the former but additionally causing congenital limb malformation [27].

*DSC3*, another cadherin gene, improves upon both tumor selectivity and safety in HNSC (AUC = 0.91 and FDR *p* = 0.065; TS safety = 8.96; and HPA safety = 1.68) and encodes the desmocollin-3 protein. As a desmosomal protein, it is mostly present in epithelial cells and is essential for the formation and adherence of desmosome cell–cell junctions. In the HPA Pathology Atlas, *DSC3* is “cancer enhanced” in head and neck cancer and cervical cancer, with the highest median expression in HNSC (94.5 FPKM) out of all cancer types. Moderate/strong cytoplasmic positivity was seen in HNSC, while DSC3 was generally lowly expressed in most other cancers [17]. While studies on this protein’s role in HNSC have been limited to the best of our knowledge, reduced expression of DSC3 in oral squamous carcinomas has been previously correlated with histological grade [28]. Biallelic inactivating mutations in *DSC3* cause skin blistering and hypotrichosis [29].

Next, *PKP3*, encoding the protein plakophilin-3, improves upon both the leading clinical tumor selectivity and safety scores for HNSC (AUC = 0.89 and FDR *p* = 0.065; TB safety = 8.67; and HPA safety = 1.55). It belongs to the plakophilin family of desmosomal proteins and links the desmosomal plaque to cytoskeletal intermediate filaments. While detected in many TCGA cancer types, it displayed the highest median expression in HNSC (82.8 FPKM). Moderate-to-high cytoplasmic immunoreactivity of this protein was yielded by the majority of HNSC cancers stained in the HPA Pathology Atlas. Intriguingly, while the upregulation of *PKP3* has been observed during the development of many cancer types, its potential tumor suppressive role has also been reported in HNSC, with its expression prognostically favorable, denoting a paradoxical role of it and other desmosomal proteins in HNSC [30]. There are no known monogenic disorders caused by mutations in *PKP3*.

The next candidate target, *COL17A1*, encodes the collagen type XVII alpha 1 chain. This gene has high tumor selectivity and safety scores for HNSC (AUC = 0.88 and FDR *p* = 0.084; TS safety = 9.13; HPA safety = 1.89). The encoded protein is part of hemidesmosomes, which are protein complexes mediating the adhesion of the dermis and epidermis. The HPA Pathology Atlas reports *COL17A1* as specifically “cancer enriched” in HNSC (164.9 FPKM), while its expression was detected in many cancer types. Moderate-to-strong cytoplasmic and membranous staining of the encoded protein was observed in HNSC samples. Collagen XVII has been found to be frequently present in HNSCs, especially those with an oral cavity or larynx origin [31]. Biallelic inactivating mutations in *COL17A1* cause intermediate junctional epidermolysis bullosa 4 [32], a genetic disorder characterized by skin blisters and erosions, nail dystrophy, and alopecia. *COL17A1* mutations can also cause epithelial current erosion dystrophy, a genetic disorder characterized by corneal erosion [33].

*MAGEA4* has higher safety scores in HNSC (TS safety = 10 and HPA safety = 1.95). It encodes the melanoma-associated antigen 4, which regulates cell proliferation by stopping cell cycle arrest at the G1 phase and decreases apoptosis mediated by p53. Importantly, the MAGE-A4 protein is already an extant CAR-T target against non-small cell lung cancer (NSCLC) [10] but has not been studied in HNSC patients. Thus, repurposing CAR therapies targeting this protein to patients with MAGEA4 + HNSC may be low cost and high yield. HPA Pathology Atlas defines *MAGEA4* to be “cancer enhanced” in lung cancer in the TCGA, with comparable expression in head and neck cancer. Medium-to-high protein expression was observed in HNSC samples by antibody staining. Its widespread expression has been previously observed in a variety of head and neck cancers [34,35,36,37,38]. There are no known monogenic disorders caused by mutations in *MAGEA4*.

We additionally explored the in vitro dependency of genes encoding known CAR targets (to our knowledge, the first systematic interrogation) and of our newly uncovered candidate CAR targets. The strong essentiality of *CD19* in a significant fraction of B-cell lymphomas and leukemias is unsurprising, given that CD19 is a B-cell marker and has an important function in B-cell receptor signaling. However, other blood tumor clinical CAR-T target genes, *CD22* and *TNFRSF17*, are not essential in any cell line of their respective indications. In solid tumors, the high essentiality of *EGFR* and *ERBB2* may suggest that these genes are highly important to the viability of cancer cells in their respective solid tumor indications. The essentiality of *IGSF3* and *SLC2A1* in a significant fraction of HNSC cell lines testifies to their importance to the viability of HNSC tumor cells. Consequently, the ability of these cancer cells to downregulate *IGSF3* and *SLC2A1* to evade CAR cell killing might be limited, upgrading their potential for serving as future CAR targets in our minds.

We note that these findings are aligned with those reported in a recent study that has characterized the dependence of the genes constituting the surfaceome in cancer cell line growth and found that the percentage of surfaceome genes defined as essential in DepMap “[was] substantially less than” the percentage of non-surfaceome genes defined as essential; only 4.1% of the former were essential versus 14.0% of the latter [7]. We note, however, that DepMap quantified in vitro gene essentiality, whose relation to in vivo essentiality observed in patients’ tumors remains to be further elucidated.

Our study has a few limitations. First and foremost, we have used gene expression as a proxy for protein expression, but it is ultimately proteins that are targeted by CARs. When advances in high-throughput single-cell proteomics technology come to the forefront in the future, it will be pertinent to apply the approach presented in this study to evaluate the discriminatory potential of membrane protein receptors as viable CAR targets. In the meantime, with the accumulating plethora of scRNA-seq datasets of tumors, gene expression is the best representation of cell genomics available to us. Second, the use of the AUC metric to evaluate target discrimination potential has its limitations. In practice, optimal discrimination relies on knowing the antigen detection thresholds of CAR-T cells. Currently, our control in setting the antigen detection density of CARs is poor, but emerging efforts may allow us to tune antigen detection cutoffs [39]. This would allow us to evaluate CAR target viability more practically. Finally, in future studies, it would be of great interest to evaluate dual- and multi-targeted CARs based on the approach outlined herewith. The computational framework enabling such combinatorial investigations has already been laid out in MadHitter [6], but for simplicity and practical applicability, we aimed to focus here on finding optimal single CAR surfaceome targets, where many such in-trial benchmarks already exist.

## 5. Conclusions

In summary, employing a systematic, data-driven analysis, we first chart the landscape of existing CAR targets in the clinic, identifying the leading targets in each cancer type and computing their estimated tumor selectivity and safety scores from pertaining single cell transcriptomics data. Next, further analyzing single-cell transcriptomics data, we performed a genome-wide search across many different cancer types to identify new and candidate CAR targets with both better selective and safety scores than those of extant clinical targets. Remarkably, in almost all cancer types, we could not find such better targets, testifying to the near optimality of the current target space, at least in accordance with our measures. However, one striking exception is HPV-negative HNSC, for which there is a dearth of existing targets in clinical trials. In the latter, our investigation has discovered 20 new and untested CAR targets for treating HNSC and GBM tumors more precisely and safely than extant targets, which are each followed up with a thorough literature survey and an analysis of their in vitro indication-specific dependency evaluation. A total of 5 of these 20 have selectivity and safety scores that are both higher than those computed for the existing CAR targets, whose further investigation may lead to better CAR treatments in HNSC.

## Figures and Tables

**Figure 1 cancers-15-04885-f001:**
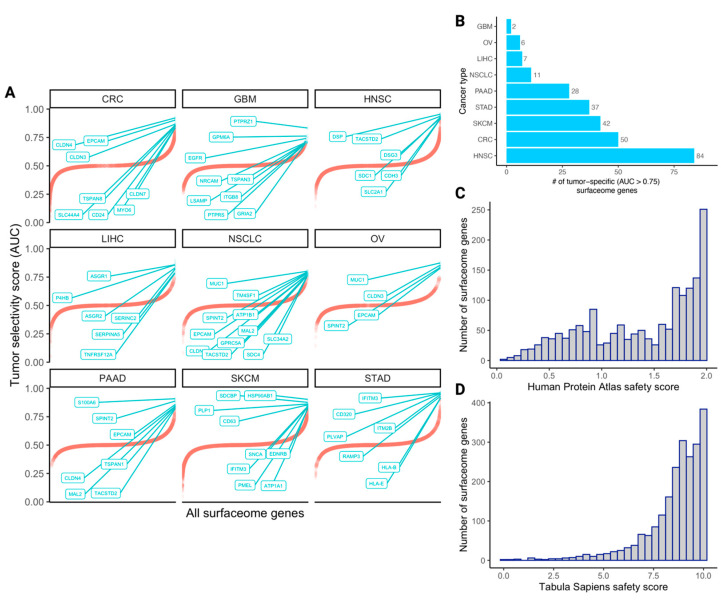
The landscape of tumor selectivity and safety scores of human surfaceome genes. (**A**) 1D scatterplots depicting the tumor selectivity scores (AUCs) of all surfaceome genes within each cancer type (red curves). Within each cancer type plot, the top N (where N is < 10) genes are labeled and represented by green points instead of red. Genes are sorted in ascending order on the x-axis from left to right. (**B**) The number of tumor-specific surfaceome genes (AUC > 0.75) in each cancer type, in ascending order from top to bottom. Histograms depicting the distributions of Human Protein Atlas (HPA)-derived safety scores and Tabula Sapiens (TS)-derived safety scores are shown in (**C**,**D**), respectively. Acronyms for the cancer types in panel A are colorectal cancer (CRC), glioblastoma (GBM), head and neck squamous cell carcinoma (HNSC), liver hepatocellular carcinoma (LIHC), non-small cell lung cancer (NSCLC), ovarian cancer (OV), pancreatic adenocarcinoma (PAAD), skin cutaneous melanoma (SKCM), and stomach adenocarcinoma (STAD).

**Figure 2 cancers-15-04885-f002:**
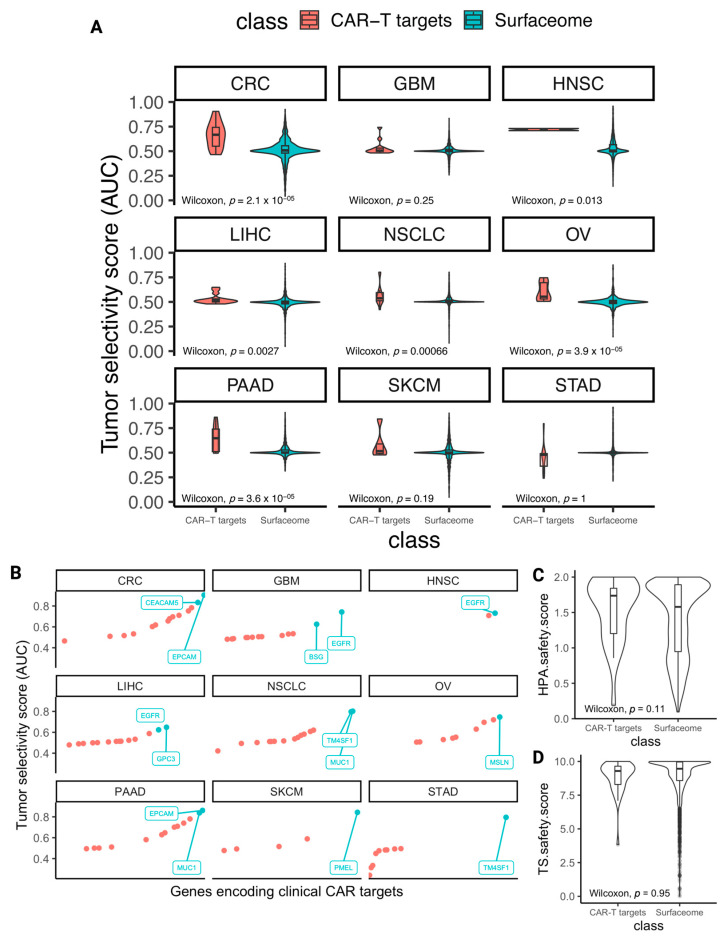
Surveying the selectivity and safety scores of known CAR targets. (**A**) Violin plots with nested boxplots depicting the tumor selectivity scores of genes encoding clinical CAR-T targets (shown in red) versus all surfaceome genes (shown in green) for each cancer type. Wilcoxon rank-sum test with *p*-values shown in each subplot, *p* < 0.05 significant. (**B**) 1D scatterplots depicting tumor selectivity scores (AUCs) of genes encoding clinical CAR targets within each cancer type. Genes whose selectivity scores are in the top 10% are represented by green points and are labeled for each cancer type plot. Genes are sorted in ascending order on the x-axis by AUC from left to right. (**C**,**D**) Distribution of safety scores of all clinical CAR target-encoding genes versus all surfaceome genes. Tabula Sapiens-derived safety scores (**C**) and HPA-derived safety scores (**D**). Wilcoxon rank-sum comparison test *p*-values are shown at the bottom of each plot. Acronyms for the cancer types in panels A and B are colorectal cancer (CRC), glioblastoma (GBM), head and neck squamous cell carcinoma (HNSC), liver hepatocellular carcinoma (LIHC), non-small cell lung cancer (NSCLC), ovarian cancer (OV), pancreatic adenocarcinoma (PAAD), skin cutaneous melanoma (SKCM), and stomach adenocarcinoma (STAD).

**Figure 3 cancers-15-04885-f003:**
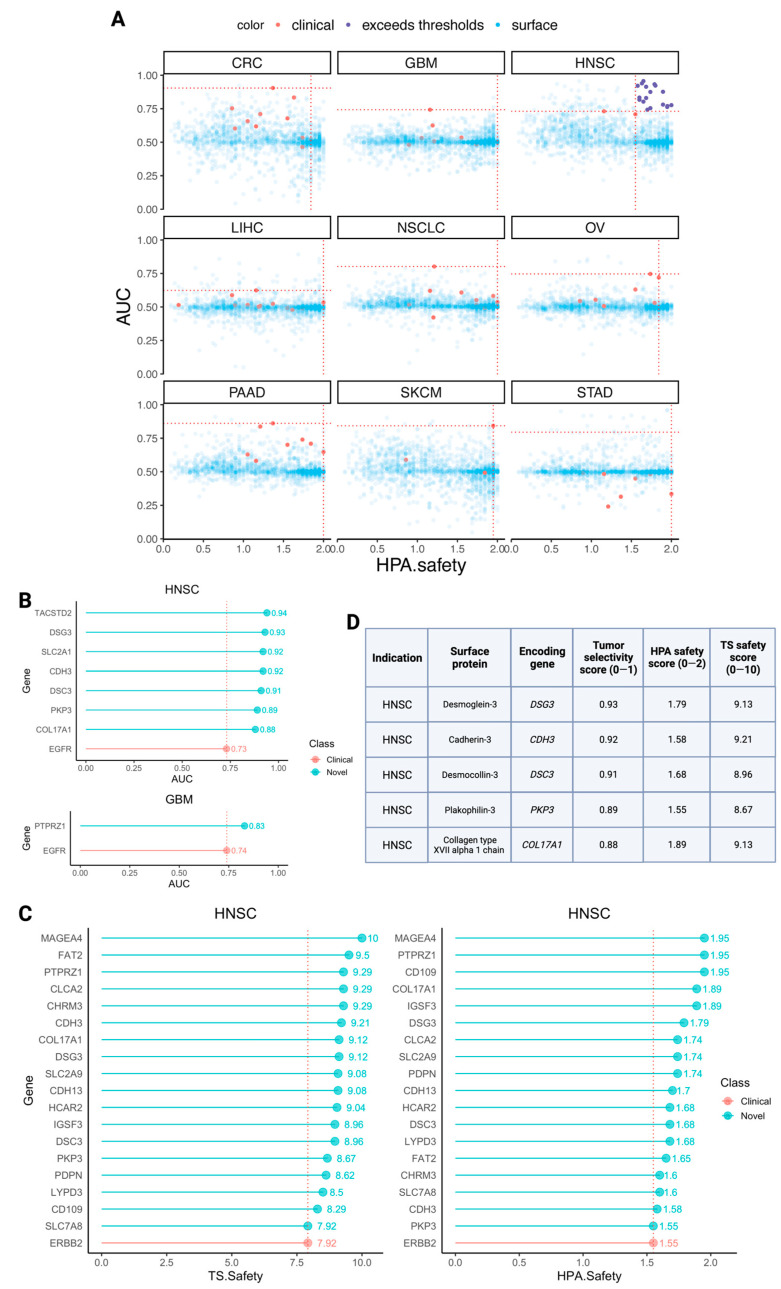
Identifications of novel CAR targets with improved selectivity and safety compared to currently leading clinical CAR targets. (**A**) Two-dimensional scatterplots of all surfaceome genes with HPA safety score on the x-axis and tumor selectivity score on the y-axis. Red points represent clinical CAR target-encoding genes for the respective cancer type. Horizontal thresholds denote the highest tumor selectivity score of clinical CAR targets pertaining to each cancer type, while vertical thresholds denote the highest respective HPA safety score of clinical CAR targets pertaining to the cancer type. Clinical CAR targets are shown in red, and surfaceome genes that surpass both the horizontal and vertical thresholds (thus appearing in the upper right quadrant) are shown in dark purple. These points emerged only for HNSC. The remaining surfaceome genes are shown in transparent blue. Tabula Sapiens safety scores were excluded for simplicity of visualization. (**B**) Top selective (and yet safe) new targets. Stems in red denote the leading clinical target by selectivity for each of the two cancer types, HNSC (left) and GBM (right), for which we found new selective targets. Stems in green denote the seven total surfaceome genes which emerged from our analysis as having greater tumor selectivity than the leading target in each cancer type. (**C**) Top safe (and yet selective) new targets. Stems colored red denote the leading clinical target by safety score, derived from HPA (left) and Tabula Sapiens (right). Stems colored green denote the surfaceome genes which emerged from our analysis as having greater safety scores than the leading clinical CAR targets derived from each atlas. Both plots depict the same set of 18 newly suggested surfaceome genes for HNSC as well as the established target gene ERBB2. (**D**) Top selective and safe new targets. New surfaceome targets appearing as both highly selective and highly safe from the previous two iterations of the analysis (shown in **B**,**C**). Rows were sorted by descending tumor selectivity scores. Acronyms for the cancer types are colorectal cancer (CRC), glioblastoma (GBM), head and neck squamous cell carcinoma (HNSC), liver hepatocellular carcinoma (LIHC), non-small cell lung cancer (NSCLC), ovarian cancer (OV), pancreatic adenocarcinoma (PAAD), skin cutaneous melanoma (SKCM), and stomach adenocarcinoma (STAD).

**Figure 4 cancers-15-04885-f004:**
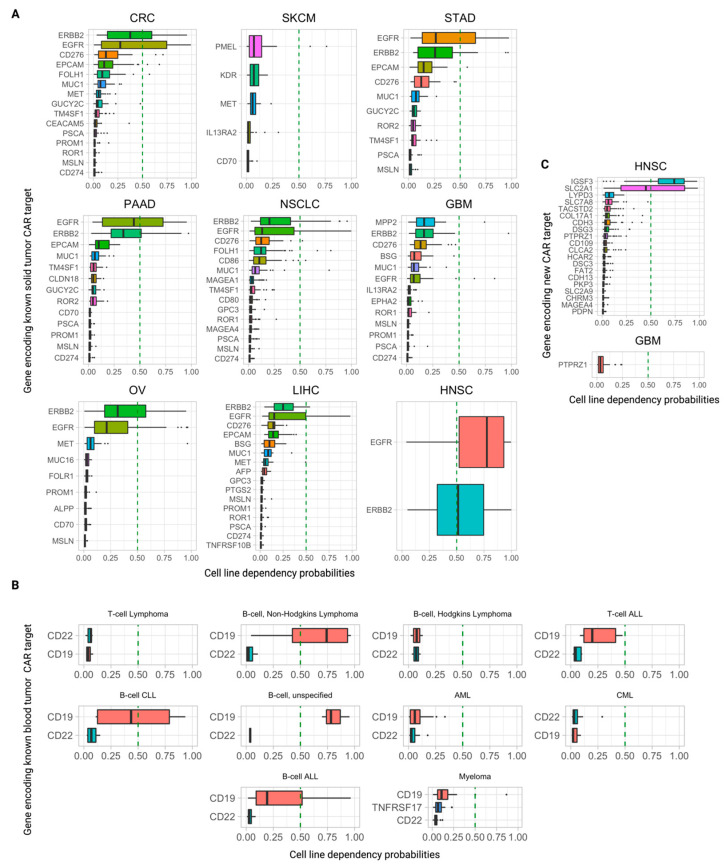
The dependency of genes encoding known CAR targets and our candidate new CAR targets in DepMap cell lines. (**A**) Probability of essentiality/dependency by cell lines on genes encoding existing solid tumor clinical CAR targets for each of the nine cancer types studied. The vertical dashed line is at p(0.50), denoting the threshold above which a cell line is considered to be “dependent” on a gene. (**B**) Probability of essentiality/dependency by cell lines on genes encoding existing clinical liquid tumor CAR targets for lymphomas, leukemias, and myeloma. The vertical dashed line is at p(0.50), denoting the threshold where a cell line is considered to be “dependent” on a gene. (**C**) Probability of essentiality/dependency by cell lines on genes encoding untested CAR targets we proposed for HNSC and GBM. The vertical dashed line is at p(0.50), denoting the threshold where a cell line is considered to be “dependent” on a gene. Acronyms for the cancer types are colorectal cancer (CRC), glioblastoma (GBM), head and neck squamous cell carcinoma (HNSC), liver hepatocellular carcinoma (LIHC), non-small cell lung cancer (NSCLC), ovarian cancer (OV), pancreatic adenocarcinoma (PAAD), skin cutaneous melanoma (SKCM), and stomach adenocarcinoma (STAD).

**Table 1 cancers-15-04885-t001:** Tissue types mapped to tissue groups in the Human Protein Atlas.

Tissue Group	Tissue Types
Adipose and soft	adipose tissue and soft tissue
Endocrine	adrenal gland, parathyroid gland, thymus, and thyroid gland
Nervous system	caudate, cerebellum, cerebral cortex, choroid plexus, hippocampus, hypothalamus, pituitary gland, substantia nigra, and dorsal raphe
Circulatory	heart muscle
Immune tissue	appendix, lymph node, spleen, and tonsil
Immune cells	bone marrow
Female tissue	breast, cervix, endometrium, fallopian tube, lactating breast, ovary, placenta, and vagina
Lungs	bronchus, lung, and nasopharynx
Colon	colon
Gastrointestinal	duodenum, oral mucosa, and rectum
Male tissue	epididymis, prostate, seminal vesicle, and testis
Esophagus	esophagus
Eye	eye, retina
Gallbladder	gallbladder
Skin	hair, skin, and sole of foot
Urinary	Kidney and urinary bladder
Liver	liver
Pancreas	pancreas
Salivary gland	salivary gland
Muscle	skeletal muscle and smooth muscle
Small intestine	small intestine
Stomach	stomach

**Table 2 cancers-15-04885-t002:** Current solid tumor CAR targets in clinics. Genes encoding cell surface proteins that are currently targeted by CARs in clinical trials for the nine solid tumor types we studied.

Cancer Type	Genes Encoding Clinical CAR Targets
CRC	*MUC1*, *CD274*, *PROM1*, *ROR1*, *PSCA*, *MET*, *EPCAM*, *EGFR*, *FOLH1*, *ERBB2*, *MSLN*, *CD276*, *TM4SF1*, *CEACAM5*, *and GUCY2C*
GBM	*MUC1*, *CD274*, *PROM1*, *ROR1*, *MPP2*, *EPHA2*, *PSCA*, *IL13RA2*, *EGFR*, *ERBB2*, *MSLN*, *CD276*, *and BSG*
LIHC	*MUC1*, *CD274*, *PROM1*, *ROR1*, *AFP*, *TNFRSF10B*, *PTGS2*, *PSCA*, *MET*, *EPCAM*, *EGFR*, *GPC3*, *ERBB2*, *MSLN*, *CD276*, *and BSG*
NSCLC	*MUC1*, *CD274*, *ROR1*, *CD80*, *CD86*, *MAGEA1*, *MAGEA4*, *PSCA*, *EGFR*, *FOLH1*, *GPC3*, *ERBB2*, *MSLN*, *CD276*, *and TM4SF1*
PAAD	*MUC1*, *CD274*, *PROM1*, *CD70*, *ROR2*, *CLDN18*, *PSCA*, *EPCAM*, *EGFR*, *ERBB2*, *MSLN*, *TM4SF1*, *and GUCY2C*
STAD	*MUC2*, *ROR2*, *PSCA*, *EPCAM*, *EGFR*, *ERBB2*, *MSLN*, *CD276*, *TM4SF1, and GUCY2C*
OV	*PROM1*, *CD70*, *FOLR1*, *MUC16*, *MET*, *EGFR*, *ERBB2*, *MSLN*, *and ALPP*
SKCM	*CD70*, *PMEL*, *KDR*, *MET*, *and IL13RA2*
HNSC	*EGFR and ERBB2*

**Table 3 cancers-15-04885-t003:** The top selectivity and safety scores of current clinical CAR targets within each cancer type.

Cancer Type	Gene with Highest Tumor Selectivity Score (0–1)	Gene with Highest HPA Safety Score (0–2)	Gene with Highest TS Safety Score (0–10)
STAD	*TM4SF1* (0.80)	*PSCA* (2.0)	*GUCY2C* (9.83)
HNSC	*EGFR* (0.73)	*ERBB2* (1.55)	*ERBB2* (7.92)
OV	*MSLN* (0.75)	*MUC16* (1.84)	*ALPP* (10.0)
SKCM	*PMEL* (0.84)	*PMEL* (1.95)	*PMEL* (9.79)
CRC	*EPCAM* (0.90)	*PSCA* (2.0)	*GUCY2C* (9.83)
GBM	*EGFR* (0.74)	*PSCA* (2.0)	*IL13RA2* (9.79)
PAAD	*EPCAM* (0.86)	*PSCA* (2.0)	*CLDN18* (9.83)
LIHC	*GPC3* (0.65)	*AFP* (2.0)	*AFP* (9.79)
NSCLC	*MUC1* (0.80)	*PSCA* (2.0)	*MAGEA1* (10.0)

**Table 4 cancers-15-04885-t004:** Families/classes of the twenty surface proteins encoded by genes identified by our analysis as candidate CAR targets.

Row	Cancer Type	Surface Protein	Encoding Gene	Protein Family/Group(s)
1	HNSC	Tumor-associated calcium signal transducer 2; Trop-2; epithelial glycoprotein-1 antigen	*TACSTD2*	Epithelial cell adhesion molecule family; transmembrane glycoprotein
2	HNSC	Desmoglein-3	*DSG3*	Desmoglein family;cadherin superfamily;transmembrane glycoprotein
3	HNSC	Cadherin-3	*CDH3*	Classical cadherin;cadherin superfamily
4	HNSC	Glucose transporter 1; solute carrier family 2	*SLC2A1*	Solute carrier 2A family
5	HNSC	Desmocollin 3	*DSC3*	Desmosomal cadherin; cadherin superfamily
6	HNSC	Plakophilin 3	*PKP3*	Plakophilin family;arm-repeat (armadillo) family; anddesmosomal protein
7	HNSC	Collagen type XVII alpha 1 chain	*COL17A1*	Collagen family; hemidesmosomal protein
8	HNSC, GBM	Protein Tyrosine Phosphatase Receptor Type Z1	*PTPRZ1*	Receptor-type tyrosine-protein phosphatase family
9	HNSC	Melanoma-associated antigen 4	*MAGEA4*	MAGE family of tumor-associated antigens
10	HNSC	FAT Atypical Cadherin 2	*FAT2*	Protocadherin;FAT family
11	HNSC	Chloride Channel Accessory 2	*CLCA2*	Calcium-activated chloride channel (CLCA) family
12	HNSC	Cholinergic Receptor Muscarinic 3; M3 Muscarinic receptor	*CHRM3*	Muscarinic acetylcholine receptors family
13	HNSC	Solute carrier family 2 member 9	*SLC2A9*	Solute carrier 2A family
14	HNSC	Cadherin-13	*CDH13*	Non-classical cadherin; cadherin superfamily
15	HNSC	Hydroxycarboxylic Acid Receptor 2	*HCAR2*	Hydroxycarboxylic acid receptor family;G-protein coupled receptor 1 family
16	HNSC	Immunoglobulin Superfamily Member 3	*IGSF3*	Immunoglobulin superfamily
17	HNSC	Podoplanin	*PDPN*	Podoplanin family; transmembrane receptor glycoprotein
18	HNSC	LY6/PLAUR Domain Containing 3	*LYPD3*	LY6/PLAUR domain-containing family;GPI-anchored metastasis-associated protein
19	HNSC	Cluster of differentiation 109	*CD109*	Alpha-2-macroglobulin family
20	HNSC	Solute carrier family 7 member 8	*SLC7A8*	Solute carrier 7A family

## Data Availability

All data analyzed in this study are publicly available via their respective cited sources. TISCH single-cell RNA-seq data were downloaded from http://tisch1.comp-genomics.org/. HPA data were downloaded from https://www.proteinatlas.org/about/download. Tabula Sapiens data were downloaded from https://figshare.com/projects/Tabula_Sapiens/100973. CRISPR DepMap data were downloaded from https://figshare.com/articles/dataset/DepMap_22Q2_Public/19700056/2.

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
