# Peer review of "Pan-Cancer Analysis of Patient Tumor Single-Cell Transcriptomes Identifies Promising Selective and Safe Chimeric Antigen Receptor Targets in Head and Neck Cancer"

_cancers, 2023, doi:10.3390/cancers15194885_

Round 1

Reviewer 1 Report

The manuscript has the putative aim of studying selective CAR-dependent targets in cells of different types of carcinomas. By analyzing the transcriptomic data of single tumor cells, the authors looked for new targets that were better than those already known. In HPV-negative head and neck squamous cell carcinoma, the authors identify twenty new candidate CAR targets.

From this reviewer's perspective, the manuscript reflects all of the journal's criteria for publication. I would suggest reviewing the text to identify small errors and clarifying the acronyms of the tables/figures.

Author Response

We thank reviewer 1 for this positive assessment. We made the following minor changes in response to the comments from reviewers.

Line 18: Inserted “of” to make the sentence structure more parallel.

Line 22: Changed “clinical CAR targets” to “CAR targets in clinical trials” to improve clarity

Line 25: Removed one of two occurrences of the header  “Abstract”

Line 28: Put selective in italics and rephrased to make clearer that what follows is an informal definition of selective.

Line 35: Spelled out “human papilloma virus” for the acronym HPV, as requested by Reviewer 1.

Lines 46-47: Spelled out the acronym “CAR-T” in the caption of Figure 1 as requested in general terms by Reviewer 1.

Line 66: Inserted “of” in “of phase II” to improve clarity.

Line 69: Spelled out Food and Drug Administration as having the acronym FDA, as requested by Reviewer 1.

Line 103: Put “Methods” in bold to make clear that this is a cross-reference to the Method section.

Line 130: Inserted “their” in “their targets, to indicate that a target is associated with a clinical trial.

Lines 131-132: Expanded the description of reference 10 to clarify how their list of clinical targets compares to our list.

Line 133: Replaced “complete up-to-date” with “larger and more up-to-date“.

Line 142: Changed “on Trialtrove” to “in Trialtrove”.

Line 205: Spelled out the acronym “FDR” as “false discovery rate”, as requested by Reviewer 1.

Lines 214-215: Spelled out ND-L as requested by Reviewer 1.

Line 216: Spelled out M-H as requested by Reviewer 1.

Line 288: spelled out TPM as requested by Reviewer 1.

Line 309: Defined the acronym TS as short for Tabula Sapiens, as requested by Reviewer 1.

Line 314: Replaced Tabula Sapiens with the acronym TS introduced at line 308.

Line 345: Spelled out Human Protein Atlas in the caption of Figure 1 and reused the acronym TS.

Lines 347-351: As requested by reviewer 1, spelled out the acronyms of all cancer types in Figure 1.

Line 378: Added a comma after “Reassuringly.”

Line 386: Reused the acronym TS, which was introduced at line 308.

Line 407-410: As requested by reviewer 1, spelled out the acronyms of all cancer types in Figure 2.

Lines 466-469: As requested by reviewer 1, spelled out the acronyms of all cancer types in Figure 3.

Line 470: Deleted “thirdly” to improve clarity.

Lines 542-546: As requested by reviewer 1, spelled out the acronyms of all cancer types in Figure 3.

Lines 561-562: Spelled out fragments per kilobase million at the first usage of the acronym FPKM, as requested by Reviewer 1.

Line 587: Corrected a typo; “TB” was changed to the intended acronym “TS”

Reviewer 2 Report

In the manuscript, the authors develop a score system based on the single cell RNA-sequencing data of tumors and healthy tissues to determine the selectivity, safety and toxicity of different CAR targets. With this system, they chart the FDA-approved and current studied CAR targets. With the aim to identify the best CAR targets with better selectivity, better safety and less toxicity, they uncovered some new promising cell surface target.

I find the paper is organized in a proper way and the results are well described. The author performed detailed background research and all the parts are organized in a logic way. The figures are well organized and presented in an appropriate way.  I have no doubt to recommend it for publication. 

good

Author Response

We thank reviewer 2 for this positive assessment. We made the following minor changes in response to the comments from reviewers 1, 2, and 4.

Line 18: Inserted “of” to make the sentence structure more parallel.

Line 22: Changed “clinical CAR targets” to “CAR targets in clinical trials” to improve clarity

Line 25: Removed one of two occurrences of the header  “Abstract”

Line 28: Put selective in italics and rephrased to make clearer that what follows is an informal definition of selective.

Line 35: Spelled out “human papilloma virus” for the acronym HPV, as requested by Reviewer 1.

Lines 46-47: Spelled out the acronym “CAR-T” in the caption of Figure 1 as requested in general terms by Reviewer 1.

Line 66: Inserted “of” in “of phase II” to improve clarity.

Line 69: Spelled out Food and Drug Administration as having the acronym FDA, as requested by Reviewer 1.

Line 103: Put “Methods” in bold to make clear that this is a cross-reference to the Method section.

Line 130: Inserted “their” in “their targets, to indicate that a target is associated with a clinical trial.

Lines 131-132: Expanded the description of reference 10 to clarify how their list of clinical targets compares to our list.

Line 133: Replaced “complete up-to-date” with “larger and more up-to-date“.

Line 142: Changed “on Trialtrove” to “in Trialtrove”.

Line 205: Spelled out the acronym “FDR” as “false discovery rate”, as requested by Reviewer 1.

Lines 214-215: Spelled out ND-L as requested by Reviewer 1.

Line 216: Spelled out M-H as requested by Reviewer 1.

Line 288: spelled out TPM as requested by Reviewer 1.

Line 309: Defined the acronym TS as short for Tabula Sapiens, as requested by Reviewer 1.

Line 314: Replaced Tabula Sapiens with the acronym TS introduced at line 308.

Line 345: Spelled out Human Protein Atlas in the caption of Figure 1 and reused the acronym TS.

Lines 347-351: As requested by reviewer 1, spelled out the acronyms of all cancer types in Figure 1.

Line 378: Added a comma after “Reassuringly.”

Line 386: Reused the acronym TS, which was introduced at line 308.

Line 407-410: As requested by reviewer 1, spelled out the acronyms of all cancer types in Figure 2.

Lines 466-469: As requested by reviewer 1, spelled out the acronyms of all cancer types in Figure 3.

Line 470: Deleted “thirdly” to improve clarity.

Lines 542-546: As requested by reviewer 1, spelled out the acronyms of all cancer types in Figure 3.

Lines 561-562: Spelled out fragments per kilobase million at the first usage of the acronym FPKM, as requested by Reviewer 1.

Line 587: Corrected a typo; “TB” was changed to the intended acronym “TS”

Reviewer 3 Report

The paper focuses the attention on a very interesting topic, as CAR therapy represent a new and promising treatment in selected haematolymphoid neoplasms.  

This is a clear paper, with structured and solid analysis of literature data.  

The results are reproducible, and the conclusions are consistent with the thesis and argument presented. 

The conclusions are interesting and add advances in the current scientific knowledge.  

No ethical problems are found in this study  

Hight quality, only minor revisions required 

Author Response

We thank reviewer 3 for this positive assessment. We made the following minor changes in response to the comments from reviewers 1, 2, and 4.

Line 18: Inserted “of” to make the sentence structure more parallel.

Line 22: Changed “clinical CAR targets” to “CAR targets in clinical trials” to improve clarity

Line 25: Removed one of two occurrences of the header  “Abstract”

Line 28: Put selective in italics and rephrased to make clearer that what follows is an informal definition of selective.

Line 35: Spelled out “human papilloma virus” for the acronym HPV, as requested by Reviewer 1.

Lines 46-47: Spelled out the acronym “CAR-T” in the caption of Figure 1 as requested in general terms by Reviewer 1.

Line 66: Inserted “of” in “of phase II” to improve clarity.

Line 69: Spelled out Food and Drug Administration as having the acronym FDA, as requested by Reviewer 1.

Line 103: Put “Methods” in bold to make clear that this is a cross-reference to the Method section.

Line 130: Inserted “their” in “their targets, to indicate that a target is associated with a clinical trial.

Lines 131-132: Expanded the description of reference 10 to clarify how their list of clinical targets compares to our list.

Line 133: Replaced “complete up-to-date” with “larger and more up-to-date“.

Line 142: Changed “on Trialtrove” to “in Trialtrove”.

Line 205: Spelled out the acronym “FDR” as “false discovery rate”, as requested by Reviewer 1.

Lines 214-215: Spelled out ND-L as requested by Reviewer 1.

Line 216: Spelled out M-H as requested by Reviewer 1.

Line 288: spelled out TPM as requested by Reviewer 1.

Line 309: Defined the acronym TS as short for Tabula Sapiens, as requested by Reviewer 1.

Line 314: Replaced Tabula Sapiens with the acronym TS introduced at line 308.

Line 345: Spelled out Human Protein Atlas in the caption of Figure 1 and reused the acronym TS.

Lines 347-351: As requested by reviewer 1, spelled out the acronyms of all cancer types in Figure 1.

Line 378: Added a comma after “Reassuringly.”

Line 386: Reused the acronym TS, which was introduced at line 308.

Line 407-410: As requested by reviewer 1, spelled out the acronyms of all cancer types in Figure 2.

Lines 466-469: As requested by reviewer 1, spelled out the acronyms of all cancer types in Figure 3.

Line 470: Deleted “thirdly” to improve clarity.

Lines 542-546: As requested by reviewer 1, spelled out the acronyms of all cancer types in Figure 3.

Lines 561-562: Spelled out fragments per kilobase million at the first usage of the acronym FPKM, as requested by Reviewer 1.

Line 587: Corrected a typo; “TB” was changed to the intended acronym “TS”

Reviewer 4 Report

The manuscript titled " Pan-cancer analysis of patient tumor single-cell transcriptomes identifies promising selective and safe CAR targets in head and neck cancer" is well written, scientifically sound and should be accepted in its present form.

NA

Author Response

We thank reviewer 4 for the entirely positive assessment. We made some minor changes in response to comments from the other three reviewers.